# Synergistic Practicing of Rhizobacteria and Silicon Improve Salt Tolerance: Implications from Boosted Oxidative Metabolism, Nutrient Uptake, Growth and Grain Yield in Mung Bean

**DOI:** 10.3390/plants11151980

**Published:** 2022-07-29

**Authors:** Sajid Mahmood, Ihsanullah Daur, Muhammad Yasir, Muhammad Waqas, Heribert Hirt

**Affiliations:** 1Department of Arid Land Agriculture, King Abdulaziz University, Jeddah 21589, Saudi Arabia; smwatto@yahoo.com (S.M.); idaur@aup.edu.pk (I.D.); 2Nuclear Institute for Agriculture and Biology (NIAB), Faisalabad 38000, Pakistan; 3Department of Agronomy, University of Agriculture Peshawar, Peshawar 25120, Pakistan; 4King Fahd Medical Research Center, King Abdulaziz University, Jeddah 21589, Saudi Arabia; yasirphr@gmail.com; 5Department of Soil Science and Plant Nutrition, Hochschule Geisenheim University, 65366 Geisenheim, Germany; wiki1466@gmail.com; 6Center for Desert Agriculture, King Abdullah University of Science and Technology, Thuwal 23955, Saudi Arabia

**Keywords:** PGPR, silicon, oxidative stress, mineral uptake, mung bean

## Abstract

Plant growth promoting rhizobacteria (PGPR) and silicon (Si) are known for alleviating abiotic stresses in crop plants. In this study, *Bacillus drentensis* and *Enterobacter cloacae* strains of PGPR and foliar application of Si were tested for regulating the antioxidant metabolism and nutrient uptake on grain yield of mung bean under irrigation of saline water (3.12 and 7.81 dS m^−1^). Bacterial inoculation and supplemental Si (1 and 2 kg ha^−1^) reduced salinity-induced oxidative stress in mung bean leaves. The improved salt stress tolerance was achieved by enhancing the activities of catalase (45%), peroxidase (43%) and ascorbate peroxidase (48%), while decreasing malondialdehyde levels (57%). Enhanced nutrient uptake of magnesium 1.85 mg g^−1^, iron 7 mg kg^−1^, zinc 49.66 mg kg^−1^ and copper 12.92 mg kg^−1^ in mung bean seeds was observed with foliar application of Si and PGPR inoculation. Biomass (7.75 t ha^−1^), number of pods per plant (16.02) and 1000 seed weight (60.95 g) of plants treated with 2 kg Si ha^−1^ and *B. drentensis* clearly outperformed treatments with Si or PGPR alone. In conclusion, application of Si and PGPR enhances mung bean productivity under saline conditions, thereby helping exploitation of agriculture in low productive areas.

## 1. Introduction

Currently, the world population reaches ~7.77 billion and will reach 9.6 billion in 2050. This fast population growth demands an increase in food production of up to 70% to feed the future world. However, arable land per capita is decreasing due to urbanization and land degradation caused by various environmental factors. Soil salinity is one of the major factors affecting 1125 million hectares of cultivable land in the 13 most affected countries with a population of around 4 billion people [1,2]. It is predicted that the intensity and area affected by salinity will increase because of poor management practices, decreased water tables, inadequate drainage of irrigated land and global warming [3].

Salinity stress disrupts various physiological, biochemical and metabolic plants functions [4]. Generally, plants in a saline rhizosphere suffer because of high osmotic stress, leading to drought as a secondary stress [5]. The excess accumulation of Na^+^ and Cl^−^ causes a nutritional imbalance [6], resulting in the accumulation of reactive oxygen species (ROS), increased lipid peroxidation and membrane damage and plant growth inhibition [7]. In response to salinity, plants try to adjust their osmotic potential to protect cells from membrane damage and ionic toxicity and to maintain water homeostasis by synthesizing osmolytes [8]. Several methods have been implemented to lower the impact of salinity stress on crop plants such as development of resistant cultivars, use of plant growth regulators and soil amendment applications [9]. Another means to alleviate salinity stress on crop plants is the use of crop supplements such as silicon and/or seed inoculation with beneficial rhizobacteria, both of which enhance salt tolerance and increase the fertility of saline soils [10,11].

Silicon has been recognized as a beneficial element for plant growth, especially under stress conditions [12]. Different studies have disclosed the positive impact of silicon on crop plants under biotic and abiotic stress [13,14]. Exogenously applied silicon could mitigate the negative effects of salinity on crop plants through inhibiting sodium uptake; regulating antioxidant metabolism; and improving gas exchange, mineral nutrition and biosynthesis of compatible solutes [15]. However, most of these experiments were performed on seedlings under hydroponic or sand culture conditions.

Inoculating seeds with beneficial rhizobacteria has received great attention in agriculture to improve crop productivity from marginal lands [16]. Microbially mediated improvement of salt tolerance has been described in different important agricultural crops including wheat, barley, rice and canola [11,17]. Rhizobacteria alleviate the adversities of salt stress via the production of phytohormones, restricting Na^+^ uptake by synthesis of exopolysaccharides, ACC-deaminase activity and the solubilization of phosphorus [18]. Individual effects of plant growth promoting rhizobacteria (PGPR) and silicon for improving salt tolerance have been reported on various agricultural crops [11,13]. However, scarce information is available on the combined application of Si and PGPR to alleviate salinity stress in crop plants [19,20]. Mung bean, an important legume crop cultivated in Asian countries, is a good source of protein and carbohydrates [21]. The majority of legume crops are sensitive to salinity stress [22], including mung bean, which fails to develop grains in the pods at electrical conductivity of 6 dS m^−1^ [23]. Cultivation of mung bean on saline soils is the major cause of low productivity in developing countries [24]. Mung bean cultivation on such lands could be profitable only if integrated with ecofriendly and sustainable approaches. It was observed in a previous study [19] that combined application of *B. drentensis* with 2 kg Si ha^−1^ resulted in the greatest enhancement of mung bean physiology, growth and yield under saline irrigation conditions. Likewise, in another study [20], the results showed that compound application of Si and PGPR countered the adverse effects of salinity on mung bean by regulating osmolytes, reducing lignification in leaves, increasing the total soluble sugars, improving mineral uptake (K^+^, Ca^2+^ and Si) and decreasing tissue Na^+^ content in comparison to the control plants and plants treated with Si and PGPR separately. Therefore, the present study was planned to explore the combined effect of Si and PGPR on improving the oxidative metabolism, micronutrient uptake, growth and grain yield of salt-stressed mung bean under natural field conditions.

## 2. Results

### 2.1. Combined Si and PGPR Treatment Positively Affects Plant Growth and Yield under Saline Conditions

Saline water irrigation significantly reduced the growth- and yield-contributing attributes in mung bean including fresh biomass, number of branches, number of pods per plant and thousand seed weight. Nevertheless, these traits were markedly improved by rhizobacterial inoculation and Si foliar application relatively to the control (Table 1). The fresh biomass significantly increased up to 28 t ha^−1^ as compared to the control, when mung bean plants were supplemented with Si (2 kg ha^−1^) in combination with *B. drentensis* inoculation under saline irrigation (3.12 dS m^−1^ EC). Among all tested treatments, combined application of Si foliar spray (2 kg ha^−1^) and *B. drentensis* inoculation produced substantially higher biomass up to 7.75 t ha^−1^ at 7.81 dS m^−1^ salinity than in single-treated plants. Foliar spray of Si without inoculations resulted in non-significant increment in fresh biomass relative to the control under both saline water treatments (3.12 and 7.81 dS m^−1^). Plants treated with PGPR (*E. cloacae* or *B. drentensis*) inoculation solely or in combination with Si under 3.12 or 7.81 dS m^−1^ salinity levels increased significantly the number of branches per plant relative to the control (Table 1). However, the maximum numbers of branches were recorded by treatment with the combination of *B. drentensis* along with Si at 2 kg ha^−1^.

Number of pods per plant remained unaffected with either bacterial inoculation or Si application alone under the salinity level of 3.12 dS m^−1^ relative to the control (Table 1). In contrast, when combined with Si application (2 kg ha^−1^) *B. drentensis* inoculation resulted in a substantially higher number of pods per plant 24.42 in relation to the *E. cloacae*-treated plants 20.25 and un-inoculated control 18.18 under salinity stress (3.12 dS m^−1^). PGPR inoculation with Si (2 kg ha^−1^) resulted in a significantly higher number of pods per plant (16.02) compared to the control plants under salinity stress (7.81 dS m^−1^).

The data of 1000 seed weight of mung bean indicated a non-significant increase in response to Si applications (1 or 2 kg ha^−1^) under both salinity stress levels (3.12 or 7.81 dS m^−1^, Table 1). In contrast, bacterial inoculation or Si application (1 or 2 kg ha^−1^) resulted in a significant increase in the 1000 seed weight compared to control plants at both salinity levels. However, the maximum increase in 1000 seed weight 87.39 and 60.95 g was recorded in plants inoculated with *B. drentensis* coupled to Si foliar application under both salinity levels (2 kg ha^−1^).

### 2.2. Leaf Mineral Contents Are Enhanced by Si and PGPR Treatment

Si application and rhizobacteria inoculation showed non-significant improvement in leaf Mg contents of mung bean under salinity stress (3.12 and 7.81 dS m^−1^, Table 2). The inoculation with *B. drentensis* and Si application (2 kg ha^−1^) collectively ensued substantial higher leaf Mg contents, 5.42 and 3.27 mg g^−1^, as compared to the control under both salinity levels (3.12 or 7.81 dS m^−1^), respectively. The Zn and Fe contents were significantly increased upon application of silicon and bacterial inoculation at both salinity levels in comparison to the control (Table 2). However, maximum Zn (184.75 and 119.90 mg kg^−1^) and Fe (356.07 and 310.79 mg kg^−1^) contents were achieved when Si application (2 kg ha^−1^) was coupled with *B. drentensis* inoculation compared to the control under both salinity levels 3.12 or 7.81 dS m^−1^, respectively.

Copper content of mung bean leaves remained unaffected in response to both salinity levels with Si applications (1 or 2 kg ha^−1^) in relation to the control (Table 2). Inoculation with both PGPR in combination with foliar Si application (1 or 2 kg ha^−1^) produced significant higher Cu contents in mung bean leaves under both salinity levels as compared to control. The combination of *B. drentensis* inoculation and Si at the rate of 2 kg ha^−1^ offered maximum increment in Cu contents up to 49.4 and 36.38 mg kg^−1^ under salinity levels of 3.12 dS m^−1^ and 7.81 dS m^−1^, respectively, as compared to the control.

### 2.3. Seed Mineral Content Is Enhanced by Si and PGPR Treatment

Irrigation with saline water adversely affected the mineral concentration of Mg, Zn, Fe and Cu in mung bean seeds. Except for Mg, bacterial inoculation and Si application resulted in substantial improvement in seed mineral content (Table 3). PGPR alone or coupled with Si application had a non-significant impact on seed Mg concentration under saline irrigation. However, Si and PGPR treatment either alone or in combination resulted in a significantly increased seed Zn concentration in comparison to non-inoculated control plants, irrespectively of Si treatment. However, maximum increment in Zn concentration, 69.36 and 49.66 mg kg^−1^, was achieved under the inoculation of *B. drentensis* with Si application (2 kg ha^−1^) under salinity levels of 3.12 dS m^−1^ and 7.81 dS m^−1^, respectively.

Foliar Si application (1 and 2 kg ha^−1^) substantially enhanced seed Fe (iron) concentration under both salinity levels in relation to control. Moreover, enhanced seed Fe concentrations, 12.82 and 7 mg kg^−1^, were observed in PGPR inoculated plants, especially when combined with foliar application of Si at 2 kg ha^−1^ at both levels of saline irrigation in comparison to control. *B. drentensis* inoculation alone or in combination with Si application under both saline irrigations resulted in substantial increases in seed Cu concentrations compared to the un-inoculated control.

### 2.4. Changes in Redox Metabolism by Si and PGPR Treatment

The application of Si enhanced the catalase (CAT) activity at salinity levels of EC 7.81 dS m^−1^, and maximum improvement was recorded when Si was applied at the rate of 2 kg ha^−1^ compared to control (Figure 1a). Inoculation with either *B. drentensis* or *E. cloacae* significantly increased the CAT activity at saline irrigation of EC 7.81 dS m^−1^ relative to the control. Furthermore, the synergistic effect of *B. drentensis* inoculation and Si foliar spray (2 kg ha^−1^) resulted in a substantially higher CAT enzyme activity (200.92 Units g^−1^ FW) among all treatments.

Significant differences were recorded for the peroxidase (POD) contents under PGPR and Si applications at both salt levels (Figure 1b). Foliar application of Si (2 kg ha^−1^) significantly increased POD levels up to 340.26 Units g^−1^ FW at the salinity level of EC 7.81 dS m^−1^ relative to the treatment without Si application, i.e., 250.66 Units g^−1^ FW. Likewise, *B. drentensis* in combination with Si (2 kg ha^−1^) under saline irrigation (EC 7.81 dS m^−1^) substantially enhanced POD levels (361.08 Units g^−1^ FW) in comparison to the control.

The application of Si on leaves (1 or 2 kg ha^−1^) markedly enhanced the ascorbate peroxidase (APX) activity at both salt levels (EC 3.12 or 7.81 dS m^−1^, Figure 1c). A substantial increase was found for APX activity under both Si applications at EC 7.81 dS m^−1^ in comparison to control plants where no Si was applied. The synergistic effect of PGPR with Si applications at both doses considerably increased APX activity levels relative to the control under salinity stress. However, the combined use of *B. drentensis* and Si at the rate of 2 kg ha^−1^ performed best among all treatments at high levels of salinity (7.81 dS m^−1^ EC).

A significant reduction in malondialdehyde (MDA) contents was observed at both doses of Si applications (1 or 2 kg ha^−1^) under both saline irrigations (3.12 and 7.81 dS m^−1^) relative to the controls (Figure 1d). Likewise, combined treatment (*B. drentensis* and *E. cloacae* inoculation and Si foliar) significantly reduced the production of MDA compared to the controls. However, more pronounced decreases in MDA contents were noted by the combined supplementation of *B. drentensis* and Si foliar application (2 kg ha^−1^) at both salinity levels (3.12 and 7.81 dS m^−1^).

### 2.5. Heat Map and Correlation Analysis of Mung Bean Traits by Si and PGPR Treatment

A heat map analyzed the complete data on the basis of treatments and then provided an instant view for the clear understanding of the trait variations using different color schemes (Figure 2). The Pearson correlation revealed the positive and negative association between all observed attributes through distinct color patterns (Figure 3). The gradual increase in the intensity of the brown color shows the strength of positive association, while the gradual increase in the intensity of the blue color shows the strength of negative association. The number of pods, number of branches, biomass and thousand grain weight are positively associated with mineral uptake and grain quality of mung bean plants while the antioxidant capacity is negatively associated with lipid peroxidation (MDA contents).

Copper content of mung bean leaves remained unaffected in response to both salinity levels with Si applications (1 or 2 kg ha^−1^) in relation to the control (Table 2). Inoculation with both PGPR in combination with foliar Si application (1 or 2 kg ha^−1^) produced significantly higher Cu contents in mung bean leaves under both salinity levels as compared to control. The combination of *B. drentensis* inoculation and Si at the rate of 2 kg ha^−1^ offered maximum increment in Cu contents up to the 49.47 and 31.71% under salinity levels of 3.12 dS m^−1^ and 7.81 dS m^−1^, respectively, as compared to the control.

Foliar Si application (1 and 2 kg ha^−1^) substantially enhanced seed Fe (iron) concentration under both salinity levels in relation to control. Moreover, enhanced seed Fe concentrations were observed in PGPR-inoculated plants, especially when combined with foliar application of Si at 2 kg ha^−1^ at both levels of saline irrigation in comparison. *B. drentensis* inoculation alone or in combination with Si application under both saline irrigations resulted in substantial increases in seed Cu concentrations compared to the un-inoculated control.

## 3. Discussion

A major cause of increased salt levels in the soil is irrigation with saline or poor-quality water, which finally hampers plant growth and yield. Although a variety of plants, called halophytes, can grow under moderate or high salt conditions, most crops belong to the glycophytes, which are salt sensitive plants. Therefore, major breeding efforts are underway to increase the salt tolerance of crops. However, other strategies to enhance salt tolerance also exist such as the use of PGPR or the treatment of plants with compounds such as Si [16]. In the present work, we assess the potential use of combining PGPR and Si to mitigate the adverse effects of salt stress on mung bean crop by saline irrigation. We report a synergistic effect of Si foliar application and rhizobacterial inoculation on salt stress tolerance that is correlated with the regulation of the antioxidative and ionic metabolism of mung beans in a 2-year field study.

Salinity perturbs various crop metabolic processes mainly by oxidative damage and ion toxicity [3,7,25], which cause severe growth reduction and yield loss [26]. As expected, salinity induced the substantial growth and yield reduction of mung bean crop in corroboration with a previous report [23], while synergistic applications of silicon and PGPR alleviated the salinity-induced damages and enhanced the performance of mung bean crop. Thus, the Si-mediated improvement in salinity tolerance of mung bean can be attributed to improved mineral nutrition, ionic homeostasis and reduction in oxidative damages by an enhanced antioxidative metabolism [27]. These findings are in accord with the previous reports, which revealed the positive effects of Si for improving the crop productivity of numerous important agricultural crops under salinity [10,13]. For instance, Si application enhanced the different organ and total biomass of sorghum crop under saline conditions [28]. Similarly, crop improvement under salinity stress by the application of bacterial inoculants has been described in various studies [11,29,30]. *Azotobacter chroococcum* (*strain C5*) inoculation of maize has the ability to process the auxin and phosphate solubilization that resulted in improvement of root and stem growth under salinity [31]. Similarly, inoculation of *Pisum sativum* by *Arthrobacter protophormiae* rhizobacteria containing the ACC deaminase significantly increased pea biomass under salt stress [22].

Nutrient imbalance and ion toxicity seriously affects plant performance, particularly under saline conditions [27]. Si and rhizobacteria applications can mitigate ion toxicity by regulating ion homeostasis under salinity stress [11,25]. Mung bean plants treated with Si and rhizobacteria inoculation accumulated high concentrations of minerals including Mg, Zn, Fe and Cu in leaves and seeds under high salinity. Our findings are in line with previous reports on cordgrass [32], tomato [33] and sorghum [25] in which Si application improved the mineral contents such as Mg, Zn, Fe and Cu. The rhizobacteria inoculation has also promoted the uptake of Mg, Ca, Fe and Mn under salinity in plants roots [11,34].

PGPR inoculation induced the improvement in mineral contents, which might be attributed to the organic acid production through bacteria by enhancing mineral availability in the soil [11]. Likewise, inoculation with PGPR enhanced Fe and Mn contents in the shoots of *Viburnum tinus* L. under salinity [34]. Moreover, Si application also had a positive effect on the population of beneficial bacteria, which promoted the availability of nutrients [35]. This might be the possible reason behind the promising role of synergetic effects of Si and rhizobacteria for increasing the productivity of mung beans in the present study.

Lipid peroxidation is an important factor for membranes damages under abiotic stresses [36], and MDA is a marker to examine ROS-induced membrane damage under stresses [37]. In this study, salinity caused membrane damages, while Si and PGPR applications lowered the oxidative damages as indicated from the lower pools of MDA in the mung bean leaves, which is in line with [38], which stated that Si augmentation reduced MDA contents in salt-stressed canola. Inoculation with *Brevibacterium linens* RS16 scaled down the MDA content in leaves of rice under salinity [39]. A strong antioxidative system of a plant is a key factor to endure adverse environmental conditions, particularly under salinity, to scavenge the overproduction of ROS and protect the cells from oxidative damages [40,41]. The results of this study revealed that activities of CAT, POD and APX in the leaf tissues were improved by supplementing Si and PGPR inoculation under salinity. In accordance with previous reports [39,41], application of Si on salt-stressed sorghum improved the activities of SOD, CAT and APX. Farshidi reported that CAT and POD activities were promoted, while APX activities were down regulated in canola by Si supplementation under salinity [38]. Likewise, promoting effects on the antioxidant activities by bacterial inoculation were reported against salinity stress, such as inoculation of *Enterobacter cloacae* HSNJ4 increasing the activities of SOD, POD and CAT under salinity in canola [17]. Moreover, *Brevibacterium linens* RS16 inoculation scaled up the SOD, CAT and APX activities in salt-stressed rice [39]. Our results show that Si and PGPR application improved the mung bean antioxidative system to reduce salinity-induced oxidative injuries (Figure 1). Thus, the strengthening the oxidative system by Si and PGPR supplementation is most likely a major factor for the improved productivity of mung beans under salinity stress.

## 4. Materials and Methods

### 4.1. Rhizobacterial Strains and Growth Conditions

Two rhizobacterial strains were obtained from the Department of Environmental Sciences (Environmental Microbiology Lab), King Abdulaziz University, Jeddah, Saudi Arabia. These strains were identified as *Bacillus drentensis* based on 16S rRNA gene sequence with accession number AJ542506 and *Enterobacter cloacae* based on MALDI-TOF spectral score > 1.9 [19]. Sterilized Luria Bertani broth medium was used for culturing the rhizobacteria and kept in a vibrating incubator for 72 h at 28 ± 1 °C and 100 rpm. The optical density was maintained 0.45 at 540 nm to adjust the microbial count 10^8^ colony forming unit mL^−1^.

### 4.2. Location and Characteristics of Site

The experiments were conducted under field conditions at Agricultural Research Station, King Abdulaziz University, Hada Al-Sham (21°48′3″ N, 39°43′25″ E), Jeddah, Saudi Arabia, for two consecutive years.

The physio-chemical properties of the soil were assessed according to Ryan [42] and are presented in Table 4. At the experimental site prevails an extreme arid climate with a mean yearly temperature of 29.22 °C and a total annual rainfall of 45 mm.

### 4.3. Details of Field Trials and Treatments

The size of the experimental plot was 3 × 2 m^2^. The treatments were arranged in a randomized complete block design including factorial settings with four repeats. Inoculation of rhizobacteria (*E. cloacae* and *B. drentensis*) was completed before sowing by mixing the mung bean seeds with a slurry containing autoclaved peat, broth culture of the specified bacterial strain and 10% sucrose solution in a ratio of 5:4:1 [23]. Treatment with slurry of the sterilized broth was considered a control treatment. Four seeds were sown per hill by maintaining proper plant-to-plant (20 cm) and row-to-row (30 cm) distance. When the seedlings had completely established, thinning was performed to one plant per hill.

The mung bean plants were irrigated with water with the two different salinities (3.12 and 7.81 dS m^−1^) daily for 10 min using automated drip irrigation system. Chemical properties of the irrigation water were examined and are presented in Table 4. The recommended dose of nitrogen (N), phosphorus (P) and potassium (K) was applied using NPK (15:15:15) compound fertilizer at the rate of 60 kg ha^−1^ at the time of sowing.

Potassium silicate used as a silicon source that was applied at either 1 or 2 kg ha^−1^ as a foliar spray in two equal splits (three weeks after sowing and at flowering stage. The known Si concentration was dissolved in water (1 L) and sprayed with a hand sprayer, while control plants were treated with the same amount of foliar application of water.

### 4.4. Measurement of the Agronomic Traits

Sampling was performed after two months of germination. Randomly, ten plants were uprooted from each plot to measure fresh biomass, number of branches, pods per plant and thousand seed weight. Meanwhile, samples for the tissue analyses were also collected and stored for further biochemical tests.

### 4.5. Antioxidants

For determining the antioxidant enzyme activities, 0.5 g fresh leaves sample from each treatment were ground using a pre-chilled pestle and mortar in 5 mL of phosphate buffer (50 mM, pH 7.8). The homogenous mixture was centrifuged (15,000× *g*) at 4 °C for 15 min. Then supernatant was collected and used for the measurements of antioxidants activities including catalase (CAT), peroxidase (POD) and ascorbate peroxidase (APX).

The activity of CAT enzyme was assessed according to the Aebi [43]. The reaction mixture of 3 mL contained the enzyme extract (100 µL), 50 mL of phosphate buffer (2.8 mL) with EDTA (2 mM, pH 7.0) and 300 mM H_2_O_2_ (100 µL). The CAT activity was examined by observing the decrease in the absorbance at 240 nm as a result of H_2_O_2_ decomposition. The CAT activity was calculated using the given formula:$$\mathrm{CAT}\;\mathrm{activity}\;\left(\mathrm{Units}/g\;\mathrm{FW}\right)=\frac{\left(\mathrm{Activity}\times A\times \frac{V}{a}\right)}{\left(E\times W\right)}$$

The POD activity was measured by following the protocol as suggested by Pütter [44]. The reaction mixture of 3 mL comprised enzyme extract (0.1 mL), 300 mM H_2_O_2_ (0.1 mL), 1.5% guaiacol (0.1 mL) and 50 mM phosphate buffer (2.7 mL, pH 7.8). The POD activity was assessed by measuring the increase in absorbance at 470 nm spectrophotometrically by the oxidation of guaiacol. The POD activity was calculated using the given formula:$$\mathrm{POD}\;\mathrm{activity}\;\left(\mathrm{Units}/g\;\mathrm{FW}\right)=\frac{\left(\mathrm{Activity}\times A\times \frac{V}{a}\right)}{\left(E\times W\right)}$$

The APX enzymes activity was measured by following the method described by Nakano and Asada [45]. The reaction mixture contained the enzyme extract (0.1 mL), 7.5 mM ascorbate (0.1 mL), 300 mM H_2_O_2_ (0.1 mL) and 50 mM phosphate buffer (2.7 mL) with EDTA (2 mM, pH 7.8). The APX oxidation was monitored by the variation in absorbance at 290 nm. The APX activity was calculated using the given formula:$$\mathrm{APX}\;\mathrm{activity}\;\left(\mathrm{Units}/g\;\mathrm{FW}\right)=\frac{\left(\mathrm{Activity}\times A\times \frac{V}{a}\right)}{\left(E\times W\right)}$$

The malondialdehyde (MDA) contents were measured by the method of thriobarbituric acid (TBA) reaction according to Heath and Packer [46] with some minor modifications [47]. Fresh leaf sample (0.25 g) was mixed with trichloro acetic acid (5 mL, 0.1% TCA). Then, the mixture was centrifuged for 15 min (12,000× *g*). The supernatant of 1 mL was mixed with 4 mL of TCA (20%) and TBA (0.5%) and was heated at 95 °C for 30 min. Then, it was cooled quickly in an ice bath and again centrifuged for 10 min (10,000× *g*). The MDA contents were calculated using the given formula:MDA (Units/g FW)=6.45×(A532 − A600) − 0.56× A450

### 4.6. Determination of Mineral Content

The leaves (dried and ground) and seed samples were analyzed for Mg, Cu, Fe and Zn contents. Briefly, 0.1 g milled samples were digested according to Wolf [48] using concentrated H_2_SO_4_ and H_2_O_2_. To the filtered and digested extract, distillated water was added to obtain the volume of 50 mL. Finally, the Mg, Cu, Fe and Zn contents were determined using inductively coupled plasma-optical emission spectrometry (ICP-OES, Varian 720-ES).

### 4.7. Data Analysis

All the data were examined using two-way ANOVA [49]. The LSD test (*p* ≤ 0.05), used for the determination of significant difference among the treatment means (*n* = 4), was performed through Statistix 8.1 (Analytical Software, Tallahassee, FL, USA), and heat map and correlation analysis were developed using MetaboAnalyst (Version 5.0).

## 5. Conclusions

In conclusion, the foliar application of Si and rhizobacteria inoculation using elite strains, which were selected from the salt-stressed field environment of Saudi Arabia, can protect mung bean from salt stress. These effects are correlated with enhanced mineral uptake (Mg, Zn, Fe, Cu), antioxidant activities (CAT, POD, APX) and reduced ROS levels in mung bean plants (Figure 4). *B. drentensis* alone or in combination with Si (2 kg ha^−1^ foliar application) exhibited more pronounced salinity mitigation action than either factor alone. Therefore, we suggest that combined application of Si and PGPR strains is a promising method to alleviate salinity damages and improve yield of crops.

## Figures and Tables

**Figure 1 plants-11-01980-f001:**
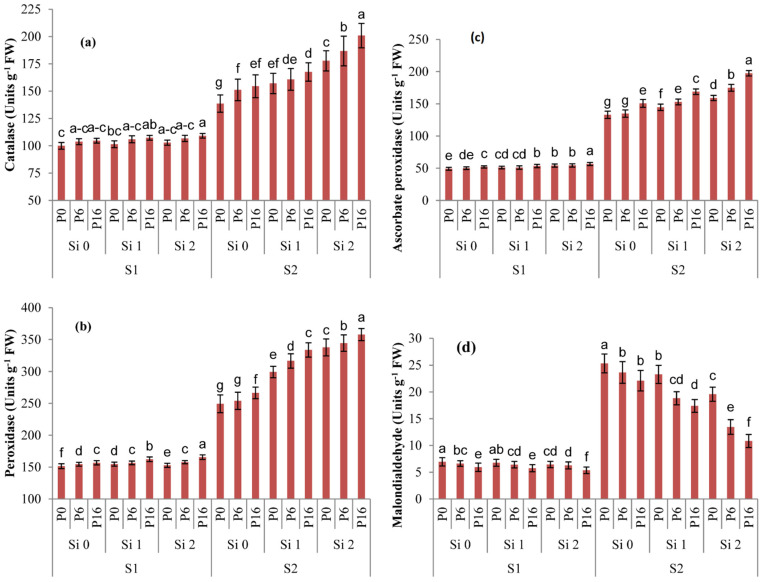
Influence of silicon (Si) and plant growth promoting bacteria (PGPR) applications on oxidative metabolism of mung bean crop under salinity stress. Oxidative metabolism was examined by antioxidants including catalase (**a**), ascorbate peroxidase (**b**), peroxidase (**c**) and lipid peroxidation level that was estimated by malondialdehyde pools (MDA) (**d**). The following are the treatment details on the x axis: P0 = Un-inoculated control, P6 = *E. cloacae,* P16 = *B. drentensis,* Si 0 = 0 kg Si ha^−1^, Si 1 = 1 kg Si ha^−1^, Si 2 = 2 kg Si ha^−1^, S1 = Saline irrigation at 3.12 dS m^−1^, S2 = Saline irrigation at 7.81 dS m^−1^. The Si was applied by foliar spray in a sequence of three weeks after sowing (1 kg ha^−1^) and at flowering stage 2 kg ha^−1^, while PGPR was applied by seed inoculation using standard protocol. Graphs represent the mean (*n* = 3) ± SE. Different letters above each bar are the indicative of statistical difference between treatments (*p* ≤ 0.05).

**Figure 2 plants-11-01980-f002:**
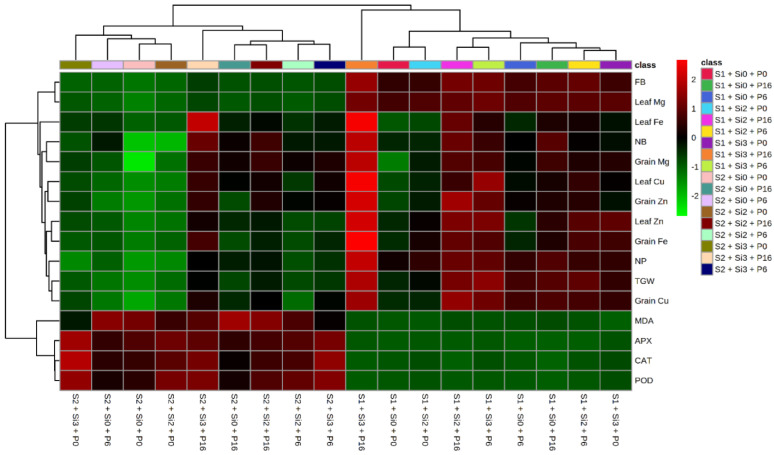
Heat map along hierarchical clustering of all observed variables in response to salinity stress with the applications of silicon (Si) and plant growth promoting bacteria (PGPR) in mung bean crop. Classes on sidebar represent the complete combinations of all set of treatments as P0 = Un-inoculated control, P6 = *E. cloacae*, P16 = *B. drentensis,* Si 0 = 0 kg Si ha^−1^, Si 1 = 1 kg Si ha^−1^, Si 2 = 2 kg Si ha^−1^, S1 = Saline irrigation at 3.12 dS m^−1^, S2 = Saline irrigation at 7.81 dS m^−1^. Side bar having gradual intensity of red and green colors shows increase or decrease in the concentration of various attributes, respectively. The complete forms of the abbreviations used for variables are FB, Fresh biomass; NP, Number of pods; NB, Number of branches; TGW, Thousand grain weight; Mg, Magnesium; Fe, Iron; Zn, Zinc; Cu, Copper; MDA, Malondialdehyde; APX, Ascorbate Peroxidase; CAT, Catalase; POD, Peroxidase.

**Figure 3 plants-11-01980-f003:**
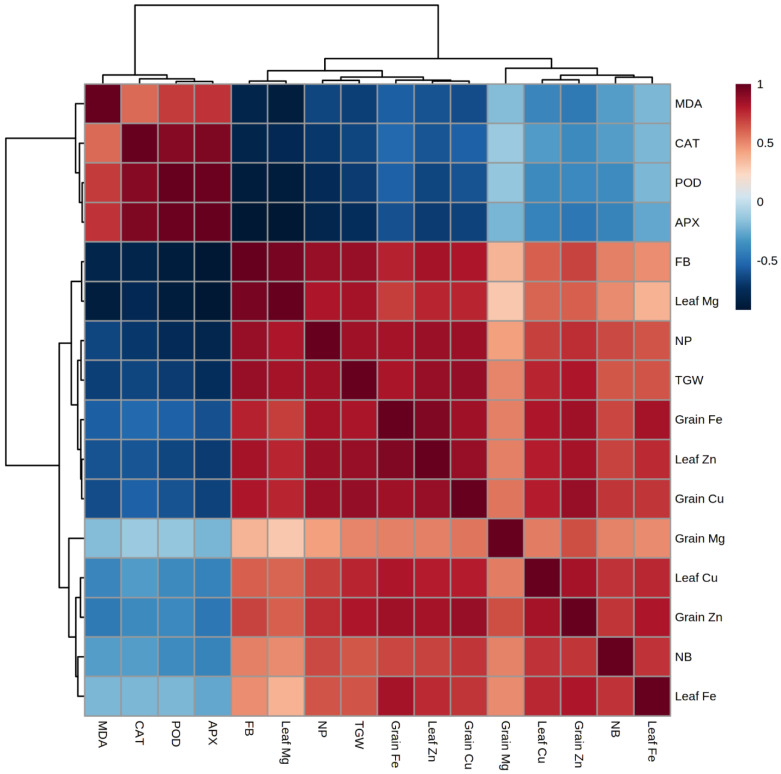
Pearson correlation with various colors showing the strength of relationship of all observed parameters. Side bar having gradual intensity of brown and blue colors depicts strength of association either positive or negative, respectively, according to the color pattern. The complete forms of the abbreviations used for variables are FB, Fresh biomass; NP, Number of pods; NB, Number of branches; TGW, Thousand grain weight; Mg, Magnesium; Fe, Iron; Zn, Zinc; Cu, Copper; MDA, Malondialdehyde; APX, Ascorbate Peroxidase; CAT, Catalase; POD, Peroxidase.

**Figure 4 plants-11-01980-f004:**
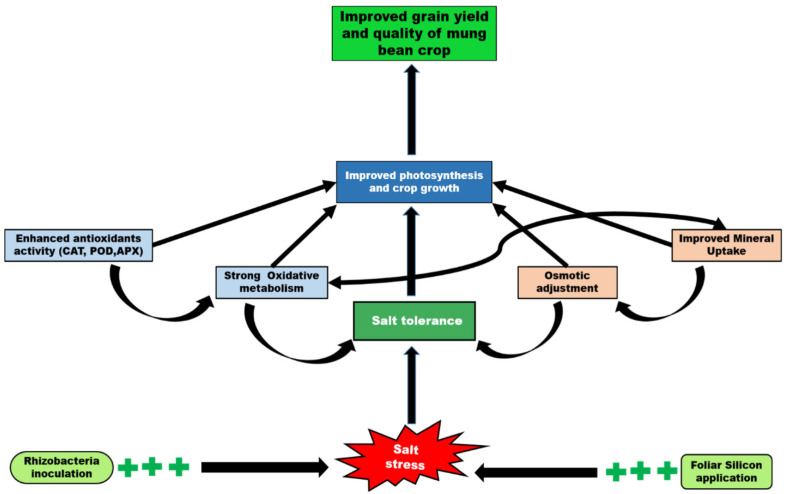
Schematic representation of possible mechanism of salt tolerance in mung bean crop induced by the synergistic effect of silicon (Si) and plant growth promoting bacteria (PGPR). The synergistic applications of Si and PGPR resulted in the improvement in numerous crop metabolisms such as oxidative metabolism through enhancement in the activity of antioxidants that, in turn, reduces the lipid peroxidation level and oxidative damages, osmoregulation by adjusting the solutes.

**Table 1 plants-11-01980-t001:** Effect of silicon foliar spray and PGPR inoculation on growth and yield of mung bean under saline irrigation conditions.

		Fresh Biomass t ha^−1^		No of Branches Plant^−1^	
Si (kg ha^−1^)	Salinity/PGPR	S1	S2	S1	S2
0	Un-inoculated	18.25 d	2.25 d	9.08 d	6.83 e
	*E. cloacae*	20.13 cd	4.13 c	9.59 cd	9.25 cd
	*B. drentensis*	22.38 bc	5.63 b	10.83 b	9.75 bc
1	Un-inoculated	18.63 d	3.75 c	9.08 d	7.00 e
	*E. cloacae*	23.25 b	5.38 b	9.67 cd	9.25 cd
	*B. drentensis*	24.75 b	5.93 b	11.08 b	10.50 ab
2	Un-inoculated	19.38 d	4.00 c	9.42 cd	8.42 d
	*E. cloacae*	24.25 b	6.13 b	10.42 bc	9.25 cd
	*B. drentensis*	28.00 a	7.75 a	12.25 a	11.08 a
	LSD	2.68	0.75	1.03	0.99
		**No of Pods Plant^−1^**		**1000 Seed Weight (g)**	
0	Un-inoculated	16.83 c	9.58 e	55.73 f	40.13 e
	*E. cloacae*	18.23 bc	12.06 d	71.38 cd	43.65 de
	*B. drentensis*	19.53 b	14.87 ab	73.99 bcd	51.00 bcd
1	Un-inoculated	18.08 bc	10.22 e	59.83 ef	45.18 cde
	*E. cloacae*	18.44 bc	12.73 cd	75.65 bcd	50.39 bcd
	*B. drentensis*	20.59 b	15.31 ab	79.20 abc	57.11 ab
2	Un-inoculated	18.18 bc	10.27 e	68.70 de	47.78 cde
	*E. cloacae*	20.25 b	13.96 bc	82.38 ab	52.88 bc
	*B. drentensis*	24.42 a	16.02 a	87.39 a	60.95 a
	LSD	2.68	1.53	8.93	7.93

Average values in the columns having different letters are significantly different at *p* ≤ 0.05 according to LSD. Where S1 = Saline irrigation at 3.12 dS m^−1^; S2 = Saline irrigation at 7.81 dS m^−1^.

**Table 2 plants-11-01980-t002:** Effect of silicon foliar spray and PGPR inoculation on mineral content of mung bean leaves under saline irrigation conditions.

		Leaf Mg (mg g^−1^)		Leaf Zn (mg kg^−1^)	
Si (kg ha^−1^)	Salinity/PGPR	S1	S2	S1	S2
0	Un-inoculated	4.86 d	2.58 c	100.35 f	68.86 g
	*E. cloacae*	5.03 b–d	2.92 abc	95.75 f	83.74 e
	*B. drentensis*	5.15 a–d	3.03 ab	128.38 d	100.76 c
1	Un-inoculated	4.99 cd	2.88 bc	116.50 e	74.92 f
	*E. cloacae*	5.14 abcd	3.01 ab	143.08 c	88.52 d
	*B. drentensis*	5.22 abc	3.16 ab	155.95 b	105.50 b
2	Un-inoculated	5.12 abcd	3.07 ab	145.63 c	88.34 d
	*E. cloacae*	5.35 ab	3.19 ab	155.50 b	91.40 d
	*B. drentensis*	5.42 a	3.27 a	184.75 a	119.90 a
	LSD	0.35	0.36	9.44	4.43
		**Leaf Fe (mg kg^−1^)**		**Leaf Cu (mg kg^−1^)**	
0	Un-inoculated	66.07 h	57.17 g	27.74 f	23.19 d
	*E. cloacae*	108.70 f	96.54 d	31.90 def	25.77 cd
	*B. drentensis*	166.24 d	113.74 c	34.24 cde	32.42 ab
1	Un-inoculated	79.80 g	64.18 f	30.34 ef	24.16 d
	*E. cloacae*	159.95 d	98.50 d	36.14 cd	28.81 bc
	*B. drentensis*	230.54 b	134.17 b	36.62 c	34.05 a
2	Un-inoculated	127.96 e	90.26 e	33.05 cde	27.62 cd
	*E. cloacae*	175.41 c	118.58 c	42.93 b	33.78 a
	*B. drentensis*	356.07 a	310.79 a	49.40 a	36.38 a
	LSD	8.88	5.89	4.27	4.54

Average values in the columns having different letters are significantly different at *p* ≤ 0.05 according to LSD. Where S1 = Saline irrigation at 3.12 dS m^−1^; S2 = Saline irrigation at 7.81 dS m^−1^.

**Table 3 plants-11-01980-t003:** Effect of silicon foliar spray and PGPR inoculation on mineral content of mung bean seed under saline irrigation conditions.

		Seed Mg (mg g^−1^)		Seed Zn (mg kg^−1^)	
Si (kg ha^−1^)	Salinity/PGPR	S1	S2	S1	S2
0	Un-inoculated	1.53 c	1.34 b	34.99 h	25.06 f
	*E. cloacae*	1.74 bc	1.60 ab	44.46 ef	28.20 ef
	*B. drentensis*	1.86 ab	1.78 a	47.02 de	34.36 d
1	Un-inoculated	1.70 bc	1.55 ab	39.97 g	29.55 e
	*E. cloacae*	1.80 b	1.77 a	48.20 d	42.73 c
	*B. drentensis*	1.90 ab	1.85 a	62.92 b	47.45 ab
2	Un-inoculated	1.81 ab	1.64 ab	41.40 fg	35.38 d
	*E. cloacae*	1.88 ab	1.81 a	55.73 c	44.21 bc
	*B. drentensis*	2.08 a	1.85 a	69.36 a	49.66 a
	LSD	0.27	0.35	3.15	4.40
		**Seed Fe (mg kg^−1^)**		**Seed Cu (mg kg^−1^)**	
0	Un-inoculated	3.47 e	0.96 e	9.31 d	3.14 f
	*E. cloacae*	3.69 e	2.21 cd	14.64 c	5.55 e
	*B. drentensis*	5.82 cd	2.52 c	15.29 bc	9.44 c
1	Un-inoculated	5.55 d	1.76 d	9.57 d	5.48 e
	*E. cloacae*	7.09 bc	2.50 c	14.87 c	6.11 e
	*B. drentensis*	7.93 b	3.86 b	18.73 a	11.40 b
2	Un-inoculated	6.82 bcd	2.05 cd	14.01 c	8.02 d
	*E. cloacae*	7.38 b	3.69 b	16.96 b	11.17 b
	*B. drentensis*	12.82 a	7.00 a	19.28 a	12.92 a
	LSD	1.45	0.48	1.69	1.15

Average values in the columns having different letters are significantly different at *p* ≤ 0.05 according to LSD. Where S1 = Saline irrigation at 3.12 dS m^−1^; S2 = Saline irrigation at 7.81 dS m^−1^.

**Table 4 plants-11-01980-t004:** Physical and chemical properties soil and water available at Hada Alsham.

	Soil	Water	
		Type I	Type II
Soil texture	Sandy loam	--	--
Organic matter (%)	0.63	--	--
Total N (%)	0.037	--	--
EC (dS m^−1^)	2.78	3.12	7.81
pH	7.73	7.90	7.93
SAR	--	6.94	16.1
	mg kg^−1^	mg L^−1^	
Na^+^	--	288.5	740.2
Cl^−^	--	74.1	165.6
P	5.8	0.14	0.13
K^+^	95	0.21	0.27
Ca^2+^	--	66.64	48.84
Mg^2+^	--	38.01	66.48
TDS	--	1983	6247
Zn^2+^	2.97	--	--
Fe^2+^	1.85	--	--
Cu^2+^	1.33	--	--

## Data Availability

Not applicable.

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
