# Peer review of "Synergistic Practicing of Rhizobacteria and Silicon Improve Salt Tolerance: Implications from Boosted Oxidative Metabolism, Nutrient Uptake, Growth and Grain Yield in Mung Bean"

_plants, 2022, doi:10.3390/plants11151980_

Round 1

Reviewer 1 Report

The article “Synergistic Practicing of Rhizobacteria and Silicon Improve Salt Tolerance: Implications from Boosted Oxidative Metabolism, Nutrient Uptake, Growth and Grain Yield in Mung bean” presents a subject of plant research relevance and explores the literature of the area. The title and subject of the manuscript are very interesting from the methodological and practical point of view, suitable and adequate the abstract of the paper is factual concrete, realistic, understandable, self-readable. However the manuscript should be improve.

  1. The current study have already reported by Mahmood, S. et al using same Bacillus drentensis and Enterobacter cloacae with same experimental setup and field condition. What is the objective of the author to repeat the same study with slight modification?
  2. The manuscript required English language editing.
  3. In abstract the author should write the results value, how much or % increase or decrease in nutrient uptake, biomass and antioxidant activities.
  4. The introduction need more relative information regarding the mung bean and the role of rhizobacteria along with silicon.
  5. Line 29-30 (Currently, the world population reaches ~7.77 billion and will reach 9.6 billion in 2050) please provide proper reference.
  6. Line 46-47: author mention the (crop supplements like silicon and/or seed inoculation with beneficial rhizobacteria). is author mean to Seed priming with rhizobacteria, if yes please mention any reference or provide more details how to process Seed priming with
  7. Author should add little information of mung bean productivity under salinity stress and how much annual loss occur due to salinity stress and epically in Saudi Arabia?
  8. Line 65-67 the object of the current study is not well clearly define.
  9. The material and method didn’t provide detail information, author should improve the material method to add more details.
  10. In material and method, the identification method is missing.
  11. Did the author submit the sequence to NCBI or gene bank, Please add the accession no of bacterial isolates.
  12. Did the author screen bacterial isolate for salinity stress?
  13. Author should check the characteristics and biochemical properties of isolates. Is the isolate have any phytohormonal activity, EPS and siderophore activities under saline condition?
  14. What concentration of Nacl did the author used? Please mention in material and method.
  15. Please provide the calculation formula of antioxidants enzymes
  16. Please correct H2O2 and H2SO4 to subscript (H2O2and H2SO4) in all the manuscript
  17. The results is fine but authors should write results in detail by mention the numerical value or percentage vale, that how much increase or decrease in agronomic traits and mineral composition in bacterial silicon or combine application of both (bacteria+SI) inoculated plants under normal and stress condition.
  18. Discussion is OK and explain well.
  19. However in line 215 what dose author refer to ‘with compoundssuch as Si16’ and line 231 ‘under saline conditions 25’.
  20. The figure quality is so poor, author should provide more clearly quality of all figures.

Reviewer 2 Report

I find the manuscript rather linear. I am not sure it is enough work for the IF of Plants but its good and neat work - well represented in clear figures. The paper provides some data of interest regarding the description of Si/PGPB effects as alleviative agents during the salinity stress and their synergistic effect on antioxidative defence, nutrient content and yield, but the mechanisms behind the observed effects are lacking.

In the MS authors presented elemental analysis of some divalent cations important for nutritional value of seeds, in seeds and leaves. Under salt stress, excessive Na+ often leads to K+ and Ca++ deficiency which is of the particular importance for plant growth. Did you analyse those elements too?

Table 2 and 3. Values for control plants grown under optimal conditions, (-salt) and without Si/PGPB addition, are not presented.

In the introduction, it should be mentioned why the mung bean was chosen, how important it is for agriculture in the region.

Round 2

Reviewer 1 Report

The article “Synergistic Practicing of Rhizobacteria and Silicon Improve Salt Tolerance: Implications from Boosted Oxidative Metabolism, Nutrient Uptake, Growth and Grain Yield in Mung bean” presents a subject of plant research relevance and explores the literature of the area.

The author have response all the comment in an appropriate manner and have made the all changes according to the current comments. The manuscript is improve well in the current form.

However in figure 1a (Catalase) the statistical variation (different latter’s) need to be correct, there is a (-) sign between the latter’s like (a-c) on some bars.

Reviewer 2 Report

After the first review, the authors responded to all the requests of the review and significantly improved the manuscript. If the editor estimates that the manuscript corresponds to the issue of the journal, it can be published in the existing form.

Author Response

Thanks anonymous reviewer for valuable comments